# Development of a Ready-to-Use Oxyresveratrol-Enriched Extract from *Artocarpus lakoocha* Roxb. Using Greener Solvents and Deep Eutectic Solvents for a Whitening Agent

Krittanon Saesue [1], Pornnapa Thanomrak [1], Wipawan Prompan [1], Warakhim Punan [1], Nantaka Khorana [1,2], Wasinee Juprasert [3], Tammanoon Rungsang [1] 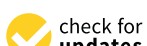, Pattravee Thong-on [4] and Jukkarin Srivilai [1,*]

1 Research and Innovation Center in Cosmetic Sciences and Natural Products, Department of Cosmetic Sciences, School of Pharmaceutical Sciences, University of Phayao, Phayao 56000, Thailand; krittanon.saesue@gmail.com (K.S.); thanomrak2542@gmail.com (P.T.); wipawanprompan@gmail.com (W.P.); kimwarakim@gmail.com (W.P.); nantaka@hotmail.com (N.K.); tammanoon.ru@up.ac.th (T.R.)
2 Faculty of Pharmacy, Payap University, Chaing Mai 50000, Thailand
3 Phitsanuchemical Co., Ltd., Phitsanulok 65000, Thailand; phitsanu-chem@outlook.co.th
4 Expert Centre of Innovative Herbal Products, Thailand Institute of Scientific and Technological Research (TISTR), Phathum Thani 10120, Thailand; pattravee@tistr.or.th
* Correspondence: jukkarin.sr@up.ac.th or jukkarint@hotmail.com; Tel.: +66-65-9419787

**Abstract:** Oxyresveratrol (ORV) is naturally found in *Artocapus lakoocha* Roxb. (AL), similar to resveratrol. This AL extract has demonstrated considerable importance in dietary supplements and cosmetics for its anti-tyrosinase and antioxidant properties. There is a great demand for ORV in the cosmetic and pharmaceutical industries. Traditionally, harsh solvents have been used to extract ORV from AL. This study aims to address this issue by introducing green technology with a ready-to-use extract for the enrichment of ORV extraction from AL using deep eutectic solvents (DESs). Thirty-three DESs were synthesized and characterized. The extraction efficiency of these DESs was evaluated by ORV content (g ORV/kg dried plant) and compared with the conventional solvents, analyzed by validated HPLC. Notably, two synthesized DESs, namely choline chloride/citric acid/water (2:1:3) (DES10) and choline chloride/xylose (1:1) (DES17), showed higher ORV content than the conventional solvents and were therefore selected for optimization of extraction conditions using Box–Behnken designs, considering three variable levels: time, temperature, and water as co-solvents. Interestingly, the biological activities of ORV-enriched extracts from DES10 and DES17 were evaluated, and the results showed that they were 74-fold and 252-fold more potent than kojic acid in terms of tyrosinase inhibitory activity. DES17 was 17-fold more potent antioxidants than ascorbic acid. The morphology of AL powder before and after extraction with DESs under SEM suggested that DESs have the same mechanism as classical organic solvents. These ORV-enriched extracts can be directly incorporated into cosmetic formulations and production scales without the need to prepare a stock solution and are therefore referred to as ready-to-use extracts. This study successfully pioneered the use of DESs for environmentally friendly and highly efficient ORV extraction from AL to produce ready-to-use extracts and applications for the pharmaceutical and cosmetic industries.

**Keywords:** *Artocarpus lakoocha* Roxb.; oxyresveratrol; deep eutectic solvents; ready-to-use extract; response surface methodology

## 1. Introduction

*Artocarpus lakoocha* Roxb. (AL), a member of the Artocarpus species belonging to the Moraceae family, is a tropical tree widely distributed in South and Southeast Asia, covering regions such as Nepal, India, Sri Lanka, Myanmar, southern China, Vietnam, Thailand, Malaysia, and Indonesia. AL is an edible plant known as "Mahad" in Thailand and "Lakuchi" in India [1]. The heartwoods of AL have been traditionally used in

Thai folk medicine for various purposes, such as treating tapeworm infections, relieving skin conditions and itching, and acting as a kidney tonic [2]. There have been many reports for its biological activities, such as antiviral (for HSV-1) [3], anti-microbial [4,5], anti-malarial [6], anti-tuberculosis [7], anti-plasmodial [7], anti-atherosclerotic [8], anti-diarrhea [9], anti-diabetic [10], wound healing [11], anti-browning for peeling fruit [12], anti-inflammatory [8,9], anti-cancer [13], anti-oxidation [13,14], and skin whitening properties [15]. In addition, bioactive compounds that have skin-whitening properties in AL are mainly found as oxyresveratrol (ORV) and resveratrol, according to recent research reports [16]. When comparing the skin whitening properties of ORV, it was found that the inhibition of tyrosinase or melanogenesis of oxyresveratrol has better efficiency compared to resveratrol [17]. ORV (trans-2,3′,4,5′-tetrahydroxystilbene, Figure 1) is a stilbenoid and was identified as the main phytochemical biomarker for AL. Other phytochemical substances contained in AL were also identified, such as resveratrol, artocarpin, etc. [18]. Nowadays, the ORV in health product development is produced using conventional methods [19–21] with harmful and non-biodegradable solvents such as chloroform, dichloromethane, methanol, ethanol, and ethyl acetate. At present, environmental friendliness is a serious concern in product development [20]. Therefore, the green process technology of using new environmentally friendly solvents to achieve a high amount of ORV and ready-to-use extract is challenging and needed for further benefit. A new generation of more environmentally friendly solvent technology has been developed in the form of novel ionic liquid analogs: deep eutectic solvents (DESs). A mixture of two or more pure chemicals that, when mixed in the appropriate ratio, results in a eutectic mixture that deviates from ideal thermodynamic behavior is known as a DES. There are strong interactions between the initial components that function as hydrogen bond donors (HBDs), such as citric acid and malic acid, and hydrogen bond acceptors (HBAs), such as choline chloride and betaine [22]. DESs have been proven to have a wide range of applications, including synthesis, extraction, biocatalysis, nanomaterials, biotechnology, electrochemistry, food, cosmetics, drugs, and biofuel [23]. As a result, DESs have been successfully applied for the extraction of natural products and to enhance bioactive content, especially flavonoids and phenolic compounds [24–28]. Ultrasound-assisted extraction (UAE) has been employed with DESs for higher bioactive yields [26–29]. The advantages of DESs include the replacement of harsh solvents, as they are less volatile, more stable, more easily biodegradable, and more environmentally friendly than harsh organic solvents [22]. However, there has been no report on novel DESs in ORV extraction from AL as ready-to-use extracts for cosmetic and pharmaceutical applications. The objective of this study was to (1) screen the best DES solvent for extracting ORV from AL; (2) optimize the ultrasonic-assisted extraction procedure with the selected optimal DES as a solvent for maximizing ORV content from AL by using response surface methodology (RSM); (3) evaluate the optimum extracts via antioxidant activity by 1,1-diphenyl-2-picrylhydrazyl (DPPH) and skin whitening activity by mushroom tyrosinase inhibitory assay with optimized DES extracts; and (4) elucidate the morphology of AL powder before and after extraction by DESs and compare it with other solvents that affect the plant cell wall structure.

**Figure 1.** The structure of oxyresveratrol (trans-2,3′,4,5′-tetrahydroxystilbene).

## 2. Materials and Methods

### 2.1. Plant Materials and Chemicals

AL was collected in April 2021 from Phayao, Thailand, and was identified by botanists. Their specimens were kept in both Queen Sirikit Botanical Gardens (QSBG), Chiang Mai, Thailand, and the School of Pharmaceutical Sciences, University of Phayao, Thailand, under the same code of PHARCOSUP0012. Heartwood parts were collected, dried at over 55 °C for 2 days, and then ground. The size of the ground powder was selected in the range of 100–250 μm and kept at reduced pressure until use. The ORV (>99% purity HPLC method, Supplementary Materials) reference standard was isolated from AL by our laboratory. Ethanol, ethyl acetate, acetonitrile, dichloromethane, dimethyl sulfoxide (DMSO), and methanol (HPLC/analytical grade) were purchased from RCI Labscan (Bangkok, Thailand). Formic acid (AR-grade) was obtained from Merck (Darmstadt, Germany). Standard buffer solutions: pH 4, pH 7, and pH 10, 2,2-diphenyl-1-picrylhydrazyl (DPPH), phosphate buffer pH 6.8, L-DOPA, tyrosinase enzyme, and kojic acid were purchased from Sigma-Aldrich (St. Louis, MO, USA). The chemicals used to prepare DESs were purchased from Aladdin Biochemical Technology Co., Ltd. (Guangzhou, China).

### 2.2. Deep Eutectic Solvent Preparation and Physicochemical Characteristics

A simple heating method was employed for the DES preparation, which was modified from a previous report [30]. The eutectic solvent consists of a hydrogen bond acceptor including choline chloride, betaine, citric acid, tartaric acid, malic acid, fructose, camphor, menthol, lauric acid, and lactic acid (Table 1) and a hydrogen bond donor including ethylene glycol, 1,3-propanediol, lactic acid, malic acid, citric acid, glycolic acid, oxalic acid, p-toluene sulfonic acid, glucose, maltose, fructose, sorbitol, xylose, xylitol, urea, 1,3 propanediol, 1,2-butanediol, tartaric acid, glucose, thymol, and menthol (Table 1), which were mixed in the appropriate molar ratio in a closed vial. The mixture was heated at ~80 °C with constant stirring using a DF-101S magnetic stirrer (Yuhua Instrument Co., Ltd., Guangzhou, China) until it became a clear liquid (approximately 1–2 h). The mixture was kept at room temperature overnight to observe the phase separation and homogeneous liquid. The stable, homogeneous, and clear liquid of the eutectic solvents was further used in the study. The chemical ingredients in molar ratio and abbreviations for DES preparation are listed in Table 1. The physicochemical properties of DES were investigated using the following characteristics: (a) water miscibility; 5 g of DES was mixed with 5 g of DI water (deionized water) and stirred at 200 rpm for 2 min. The DES miscible with water was classified as hydrophilic DES, while the immiscible DES in water was classified as hydrophobic DES. (b) The pH and (c) electrical conductivity of the DESs were determined at room temperature using a S470 kit pH meter and conductivity meter (Mettler Toledo, Columbus, OH, USA). (d) The viscosity of the DESs was measured using a Brookfield DV-II + Pro viscometer (Mettler Toledo., Columbus, OH, USA) at room temperature with a CPE spindle No. 41 at 12 rpm.

**Table 1.** Groups, components, and solubility properties of DESs in this study.

| Groups | DESs No. | Hydrogen Bond Acceptor (HBA) | Hydrogen Bond Donor (HBD) | Co-Solvent | Molar Ratio (HBA/HBD/Co-Solvent) | Solubility Property | Reference |
|---|---|---|---|---|---|---|---|
| G1;ChChl-glycol | DES1 | Choline chloride | Ethylene glycol | - | 1:2 | Hydrophilic | [31] |
| | DES2 | Choline chloride | 1,3-Propanediol | - | 1:3 | Hydrophilic | [32] |
| | DES3 | Choline chloride | Lactic acid | Water | 3:1:2 | Hydrophilic | [33] |
| G2;ChChl-acid | DES4 | Choline chloride | Lactic acid | - | 1:1 | Hydrophilic | [33] |
| | DES5 | Choline chloride | Lactic acid | - | 1:2 | Hydrophilic | [33] |
| | DES6 | Choline chloride | Lactic acid | - | 1:3 | Hydrophilic | [33] |
| | DES7 | Choline chloride | Malic acid | Water | 2:1:1 | Hydrophilic | [34] |
| | DES8 | Choline chloride | Malic acid | Water | 1:1:1.2 | Hydrophilic | [34] |
| | DES9 | Choline chloride | Malic acid | Water | 1:1:3 | Hydrophilic | [34] |
| | DES10 | Choline chloride | Citric acid | Water | 2:1:3 | Hydrophilic | [35] |

**Table 1.** *Cont.*

| Groups | DESs No. | Hydrogen Bond Acceptor (HBA) | Hydrogen Bond Donor (HBD) | Co-Solvent | Molar Ratio (HBA/HBD/Co-Solvent) | Solubility Property | Reference |
|---|---|---|---|---|---|---|---|
| | DES11 | Choline chloride | Glycolic acid, Oxalic acid | - | 1:1.7:0.3 | Hydrophilic | [36] |
| | DES12 | Choline chloride | P-Toluene sulfonic acid | - | 1:1 | Hydrophilic | [37] |
| | DES13 | Choline chloride | Glucose | Water | 2:1:3 | Hydrophilic | [38] |
| | DES14 | Choline chloride | Maltose | Water | 4:1:6 | Hydrophilic | [39] |
| G3;ChChl-sugar | DES15 | Choline chloride | Fructose | - | 2:1 | Hydrophilic | [40] |
| | DES16 | Choline chloride | Sorbitol | - | 1:1 | Hydrophilic | [41] |
| | DES17 | Choline chloride | Xylose | - | 1:1 | Hydrophilic | [42] |
| | DES18 | Choline chloride | Xylitol | Water | 1:1:1.2 | Hydrophilic | [42] |
| | DES19 | Choline chloride | Xylitol | Water | 1:1:4.75 | Hydrophilic | [42] |
| G4;ChChl-urea | DES20 | Choline chloride | Urea | Water | 1:2:3 | Hydrophilic | [41] |
| G5;Betaine-glycol | DES21 | Betaine | 1,3-Propanediol | - | 1:5 | Hydrophilic | [43] |
| | DES22 | Betaine | 1,2-Butanediol | - | 1:7 | Hydrophilic | [44] |
| G6;Betaine-acid | DES23 | Betaine | Tartaric acid | Water | 1:1:1 | Hydrophilic | [45] |
| | DES24 | Citric acid | Glucose | Water | 1:1:2.75 | Hydrophilic | [46] |
| G7;Sugar-acid | DES25 | Citric acid | Xylitol | Water | 1:1:2.75 | Hydrophilic | [47] |
| | DES26 | Tartaric acid | Glucose | Water | 1:1:4.25 | Hydrophilic | [48] |
| | DES27 | Malic acid | Xylitol | Water | 1:1:1.2 | Hydrophilic | [49] |
| G8;Sugar-sugar | DES28 | Fructose | Glucose | Water | 1:1:11 | Hydrophilic | [50] |
| G9;Natural | DES29 | Honey | | - | - | Hydrophilic | [51] |
| | DES30 | Camphor | Thymol | - | 1:1 | Hydrophobic | [52] |
| G10;Hydrophobic | DES31 | Menthol | Thymol | - | 1:1 | Hydrophobic | [52] |
| | DES32 | Lauric acid | Menthol | - | 1:2 | Hydrophobic | [53] |
| | DES33 | Lactic acid | Menthol | - | 1:1 | Hydrophobic | [54] |

## 2.3. Isolation of Oxyresveratrol for High-Performance Liquid Chromatography Analysis

The oxyresveratrol was isolated from dried AL according to some modifications from a previous methodology [55]. Briefly, the powder (250 g) of AL was macerated with 95% ethanol (3 L × 3) and then filtered. The filtered solution was concentrated under reduced pressure to afford the crude extract (10% yield). The crude extract (12 g) was dissolved in methylene chloride and then fractionated using quick column chromatography (10 × 13 cm) with a gradient elution of methylene chloride and methanol in a ratio of 100:0 to 0:100. Fifteen fractions (250 mL) were collected and monitored using TLC under UV 254 and 366 nm. The fractions 7 to 10 were pooled and concentrated under reduced pressure to provide yellow crystals, and recrystallization was carried out using methanol and ethyl acetate to provide the white powder of oxyresveratrol (159 mg). The powder was characterized by a Bruker Avance 400 nuclear magnetic resonance (NMR) spectrophotometer (Bruker, Damstadt, Germany) (Figure S6a,b), Fourier transform infrared (FT-IR), IRAffinity-1S spectrophotometer (Shimadzu, Tokyo, Japan) (Figure S7) and LC-QTOF-MS/MS (Agilent 1260 coupled to an Agilent-6540 UHD QTOF mass spectrometer, Agilent Technologies, Santa Clara, CA, USA) (Figure S8). The isolated compound was then analyzed using the HPLC method to calculate purity (Figure S5). The ORV was used as a reference standard for HPLC analysis.

## 2.4. High-Performance Liquid Chromatography Analysis and Method Validation

Oxyresveratrol was analyzed using the HPLC method. The HPLC method was performed using the HPLC Shimadzu Prominence UFLC system equipped with a LC-20AD pump, an SPD-20A 230 V UV-Vis detector, a rheodyne injector with a 20 µL loop, and a 5 µm C-18(2) column (Phenomenex) with a 250 mm × 4.6 mm diameter. The sample injection volume was 20 µL, and the isocratic elution was applied at a flow rate of 1 mL/min under room temperature conditions. The mobile phase consisted of 1.0% (*v/v*) formic acid in DI water (A) and acetonitrile (B), with a ratio of A:B of 35:65. The mobile phase was filtered through a 0.45 µm filter with a vacuum filter and degassed with an ultrasonic bath for 10 min before use. The detector wavelength was set at 325 nm. For the validation of

the HPLC method, a stock solution of ORV was prepared by dissolving in ethanol, and the calibration curve was constructed at concentrations of 1.00–20.00 μg/mL. To assure the reliability and validity of HPLC analysis, the HPLC method was validated by these parameters of linearity and range, the lowest detection limit concentration (LOD), the lowest quantification limit concentration (LOQ), accuracy (%recovery), and precision (%RSD) according to Q2(R2) ICH guidelines (2022) [56].

### 2.5. Screening of Deep Eutectic Solvents for Oxyresveratrol Extraction

Firstly, all the synthesized DESs were screened at a plant-to-sample ratio of 1:80 (*w/w*), and 10 mg of dried powder of AL was added to the DES of 800 mg. The extraction temperature was set at 50 ± 2 °C, and an ultrasonic bath was held for 30 min. at 40 kHz (KQ3200DE, Kunshan Ultrasonic Instruments, Shanghai, China). The classical solvents, ethyl acetate, ethanol, and propylene glycol, were used to compare the extraction efficiency. Upon completion of the extraction period, the resulting extract was filtered through a 0.45 μm nylon filter. The filtered solution extract was subsequently diluted with 99% ethanol (for hydrophobic DESs) and 50% ethanol (for hydrophilic DESs) before undergoing analysis using a validated HPLC method. All the extracts were determined by ORV content and calculated as g ORV/kg dried weight of plant (g/kg DW) as extraction efficiency. Each extraction procedure was performed in triplicate. The extract from DESs with the maximum ORV content was then chosen for further experimentation for optimization of extraction conditions.

### 2.6. Optimization Conditions for Extraction

Box–Behnken Design (BBD) was used with 3 factors and 3 levels to find the optimal values for three independent variables: extraction time (5–120 min), temperature (30–65 °C), and water as a co-solvent (DES10 with a 3–105 molar ratio and DES17 with a 0–65 molar ratio). The process variables, including X1: extraction time (min), X2: temperature (°C), and X3: molar ratio of water in DES, were optimized, and BBD was evaluated using three levels for each variable as follows:

DES10:    (X1: 5 min (−1), 62.5 min (0), 120 min (1); X2: 35 °C (−1),
            47 °C (0), 60 °C (1); X3: 3 (−1), 54 (0), 105 (1)

DES17:    (X1: 5 min (−1), 62.5 min (0), 120 min (1); X2: 35 °C (−1),
            47 °C (0), 60 °C (1); X3: 0 (−1), 32.5 (0), 65 (1)

In this study, a total of 15 experiments were performed to optimize the extraction parameters for ORV content (g/kg DW). An analysis of variance (ANOVA) was conducted to validate the theoretical accounts of the optimization process. The estimation of optimal conditions was achieved by a second-order polynomial equation. The generalized form describes the relationship between the responses and the parameters as follows in Equation (1):

$$Y = \beta_0 + \sum_{i=1}^{n} \beta_i x_i + \sum_{i=1}^{n} \beta_{ii} x_i^2 + \sum_{i \leq 1 \leq j}^{n} \beta_{ij} x_i x_j + \varepsilon \tag{1}$$

In the equation, Y represents the response, while $\beta_0$, $\beta_i$, $\beta_{ii}$, and $\beta_{ij}$ are the regression coefficients for the intercept, linear, interaction, and quadratic, respectively. The $x_i$ and $x_j$ represent the independent variables and the number of independent parameters (*n* = 3). The analyses were performed using a trial version of Design-Expert 13 (Stat-Ease, Minneapolis, MN, USA). The model adequacy was assessed based on the obtained coefficient of multiple determination ($R^2$), coefficient of variance (CV), and *p*-values for the model and the test for lack of fit.

### 2.7. DPPH Radical Scavenging Assay

The percentage of DPPH inhibition in each sample was determined using a DPPH radical assay. The measurement of DPPH radical-scavenging activity was performed according to a previous methodology [57]. The samples reacted with the stable DPPH radical in an ethanol solution. The reaction mixture consisted of the addition of 100 μL of sample and 100 μL of a 0.3 mM solution of DPPH radical in ethanol. When DPPH

reacts with an antioxidant compound, which can donate hydrogen, it is reduced. The color changes (from deep purple to light yellow) were measured at 517 nm after 30 min of reaction time using a microplate reader (Biotek Synergy H1, Winooski, VT, USA). The mixture of ethanol (100 μL) and sample (100 μL) served as a blank. The control solution was prepared by mixing the solvent of the sample (100 μL) with the DPPH radical solution (100 μL). Each measurement was performed in triplicate. The percentage of DPPH inhibition was determined according to the following Equation (2):

$$\%\text{DPPH inhibition} = 100 - \left[ \frac{\left( \text{Abs}_{\text{sample}} - \text{Abs}_{\text{blank}} \right) \times 100}{\text{Abs}_{\text{control}}} \right] \tag{2}$$

### 2.8. Mushroom Tyrosinase Inhibition Assay

A mushroom tyrosinase inhibition assay was performed using the biochemical conversion of L-DOPA to dopachrome by tyrosinase according to a previous methodology [58]. The reaction mixture consisted of the addition of 110 μL of phosphate buffer (0.05 M, pH 6.8), 40 μL of mushroom tyrosinase (200 U/mL), and 10 μL of sample solution. After 10 min, add 40 μL of 0.85 μM L-DOPA in phosphate buffer. After 20 min of L-DOPA addition, the reaction was monitored at 490 nm for dopachrome formation in the reaction mixture. Kojic acid was used as a positive control. The concentration range of the extract used for the mushroom tyrosinase inhibition assay was 0–0.3 mg/mL. The mixture of phosphate buffer (150 μL), L-DOPA (40 μL), and sample (10 μL) served as a blank. The control solution was prepared by mixing phosphate buffer (110 μL), L-DOPA (40 μL), mushroom tyrosinase (40 μL), and solvent (10 μL). Each measurement was performed in triplicate. The percentage of tyrosinase inhibitory activity was determined according to the following Equation (3):

$$\%\text{Inhibition of tyrosinase activity} = 100 - \left[ \frac{\left( \text{Abs}_{\text{sample}} - \text{Abs}_{\text{blank}} \right) \times 100}{\text{Abs}_{\text{control}}} \right] \tag{3}$$

### 2.9. Scanning Electron Microscopy

The morphology of the dried AL powder was evaluated by scanning electron microscopy (SEM) analysis, which was adapted from a previous methodology [59]. The powder samples of AL before and after extraction with different solvents, including DES10, DES17, ethanol, ethyl acetate, and propylene glycol, were dried up at 50 °C for 24 h and then kept in a desiccator before use. The samples were placed on the sample stand, and their surfaces were then coated with a Sputter Coater or a gold-coated putty machine and examined using a scanning electron microscope (Oxford Instrument Co., Oxfordshire, UK) at a difference magnitude of 4000× and 16,000×.

### 2.10. Data Analysis

All data were expressed as mean ± SD. The screening DESs for ORV extraction results were expressed as mg ORV per kg dried weight of plant (mg/kg DW). DPPH scavenging activity and tyrosinase inhibition activity were plotted against log10 [concentration], and the half maximum inhibitory concentration (IC$_{50}$) was calculated using GraphPad Prism Software version 9 (GraphPad Software, San Digo, CA, USA). All samples were performed in triplicate. The statistical analysis of the data was conducted by one-way ANOVA at a 95% confidence level ($p < 0.05$) using the general linear model of SPSS 14.0 (IBM, New York, NY, USA). In addition, the mean and the standard deviation were all calculated, at least in triplicate, from the experiment.

## 3. Results

### 3.1. Deep Eutectic Solvent Preparation and Physicochemical Properties

Here, we attempted to comprehensively screen DESs for the extraction of ORV from AL. Therefore, 33 DESs were synthesized according to achievable systems from previous reports, and some systems were investigated by our research group (Table 1). A clear liquid in deep eutectic systems at room temperature was selected as the solvent for extraction. The obtained DESs were classified according to their composition into 10 groups (G1–G10), including G1: ChChl/glycol, G2: ChChl/acid, G3: ChChl/sugar, G4: ChChl/urea, G5: betaine/glycol, G6: betaine/acid, G7: sugar/acid, G8: sugar/sugar, G9: natural DES, and G10: hydrophobic DESs (Table 1). The water miscibility of the DESs was also determined and classified as hydrophilic if they were completely miscible with water. Hydrophobic DESs, on the other hand, were immiscible with water. The result showed that DESs 1–29 were classified as hydrophilic and DESs 30–33 as hydrophobic according to their miscibility properties in water (Figure 2).

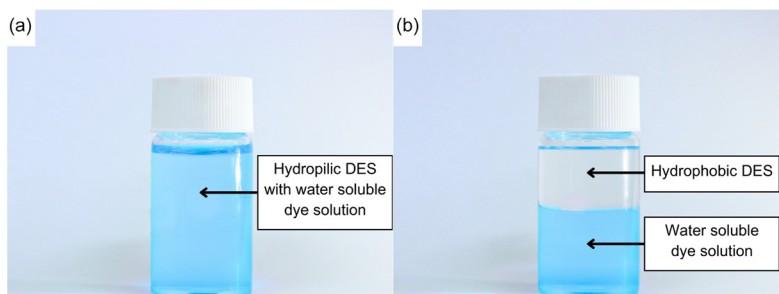

**Figure 2.** The water-soluble dye solution is miscible with DES10 as a representative of the hydrophilic DESs (**a**) and immiscible with DES32 as a representative of the hydrophobic DESs (**b**).

The pH, viscosity, and conductivity of the DESs were characterized, and the results are shown in Figure S1a, S1b and S1c, respectively. We found that G1-2 and G6-8 had a pH in the acidic range (pH 1–6). The ChChl/glycol (G1), ChChl/acid (G2), and sugar/acid (G7) had pH ranges of 2–6, 0–3, and 0–2, respectively, while the betaine/acid (G6) and sugar/sugar (G8) had pH ranges of 2–3. The neutral and alkaline ranges (pH 7–10) were mainly observed in G3–G5, including betaine/glycol (G5) and ChChl/urea (G4), with pH ranges of 7–9 and 9–10, respectively. The DESs in ChChl/sugar (G3) were mostly acidic, except for ChChl/xylitol (DES18–19), which was neutral. The viscosity of all DESs was presented in Figure S1b in the range of 0.98–4505.00 cP. ChChl/acid (G2), sugar/sugar (G8), and sugar/acid (G7) showed moderate viscosity in the range of 400–600 cP, and ChChl/sugar (G3) had high viscosity, such as DES16 (ChChl/sorbitol, 1:1) and DES17 (ChChl/xylose, 1:1) with 3576.00 ± 0.015 cP and 4505.00 ± 8.663 cP, respectively. The electrical conductivity of the DESs was mostly in the range of 100–500 mV, and some were in the negative range. The results are shown in Figure S1c.

### 3.2. High-Performance Liquid Chromatography and Method Validation for Oxyresveratrol Analysis

The results showed that the retention time of ORV was 5.12 min under the detection of 325 nm. (Figure 3a,b). The major compound, ORV, in the AL extract was observed at a similar retention time to the standard ORV in the chromatogram of the AL extract from DES10, a representative extract. Linearity was determined using five concentrations in the range of 1.00–20.00 µg/mL, suitable for ORV determination in the extract. The calibration curves (Figure S2, Table 2) revealed that the linear regression equation had good correlation coefficients ($r^2 = 0.9998$). The LOD and LOQ were calculated and confirmed by dilution determination and were 0.0025 and 0.010 µg/mL, respectively. To validate the HPLC analysis, the three concentrations from the range of analysis were evaluated as QC1, QC2, and QC3 at 4, 10, and 16 µg/mL. The accuracy of assay was evaluated by %recovery using the spiking technique while the precision of assay was assessed by %RSD using intra-day

and inter-day precision in three consecutive days. The results showed that the accuracy of assay was in the acceptable range of 98.9 to 100.3% and 100.11 to 102.33%, respectively. The intra-day and inter-day precision were in the acceptable range of 1.11 to 1.68% and 0.86 to 2.42%, respectively. The correlation coefficient (>0.9950), precision (%RSD < 5%), and recovery (80–120%) supported the suitability of the HPLC method for ORV analysis according to the ICH guidelines.

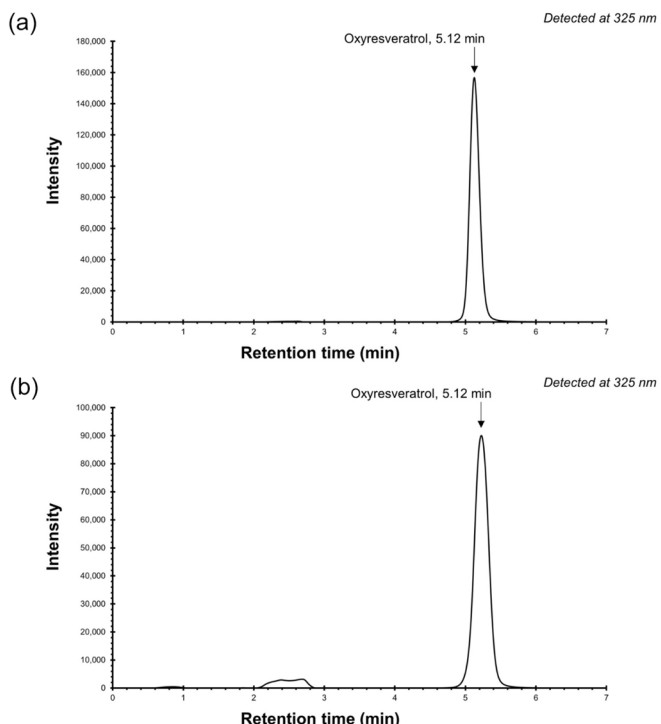

**Figure 3.** HPLC chromatograms of the reference standard of ORV (**a**) and the HPLC chromatogram of the AL extract sample of DES10 (**b**), showing the retention time of ORV at 5.12 min under the detection of 325 nm.

**Table 2.** The method validation results for HPLC analysis of ORV.

| Validation Parameter | Value |
| --- | --- |
| Linearity | 1.00–20.00 µg/mL |
| Correlation coefficient ($R^2$) | 0.9998 |
| Linear equation | y = 123,761x − 13,184 |
| LOD | 0.0025 µg/mL |
| LOQ | 0.010 µg/mL |
| Precision (%RSD); Inter day | 1.11 to 1.68% |
| Precision (%RSD); Intra day | 0.86 to 2.42% |
| Accuracy (%Recovery); Inter day | 100.11 to 102.33% |

### 3.3. Screening Deep Eutectic Solvents for Oxyresveratrol Extraction

A comparative analysis of 33-DES solvents versus classical organic solvents in the extraction of ORV from AL under the same condition using ultrasonic-assisted extraction (sample to solvent ratio of 1:80, temperature at 50 ± 2 °C for 30 min) was evaluated. The findings, depicted in Figure 4, revealed that the ORV content of the extracts with the classical solvents, ethanol, ethyl acetate, and propylene glycol, was in the range of 40–60 mg ORV/kg DW. Although natural deep eutectic solvents like honey, commonly used as a vehicle in traditional medicine, were tested, they yielded lower ORV content compared to

classical solvents, so more extraction time was suggested. Remarkably, the extracts of DESs such as ChChl/citric acid/water (DES10) in the ChChl/acid group (G2), ChChl/sorbitol (DES16), and ChChl/xylose (DES17) in the ChChl/sugar group (G3) exhibited outstanding performance, yielding ORV content in the range of 100–130 mg ORV/kg DW, which were 2–3 times higher than those of classical solvents. These results were consistent with previous findings demonstrating the efficacy of ChChl/acid in flavonoid extraction [60]. However, there has been no report on ChChl/sugar for flavonoid extraction from plants. Three promising DESs, namely DES10, DES16, and DES17, exhibited high ORV content, but only two were selected from the representatives of the ChChl/acid group (G2) and the ChChl/sugar group (G3), namely DES10 and DES17, which were selected for further optimization of the extraction conditions to maximize the ORV content.

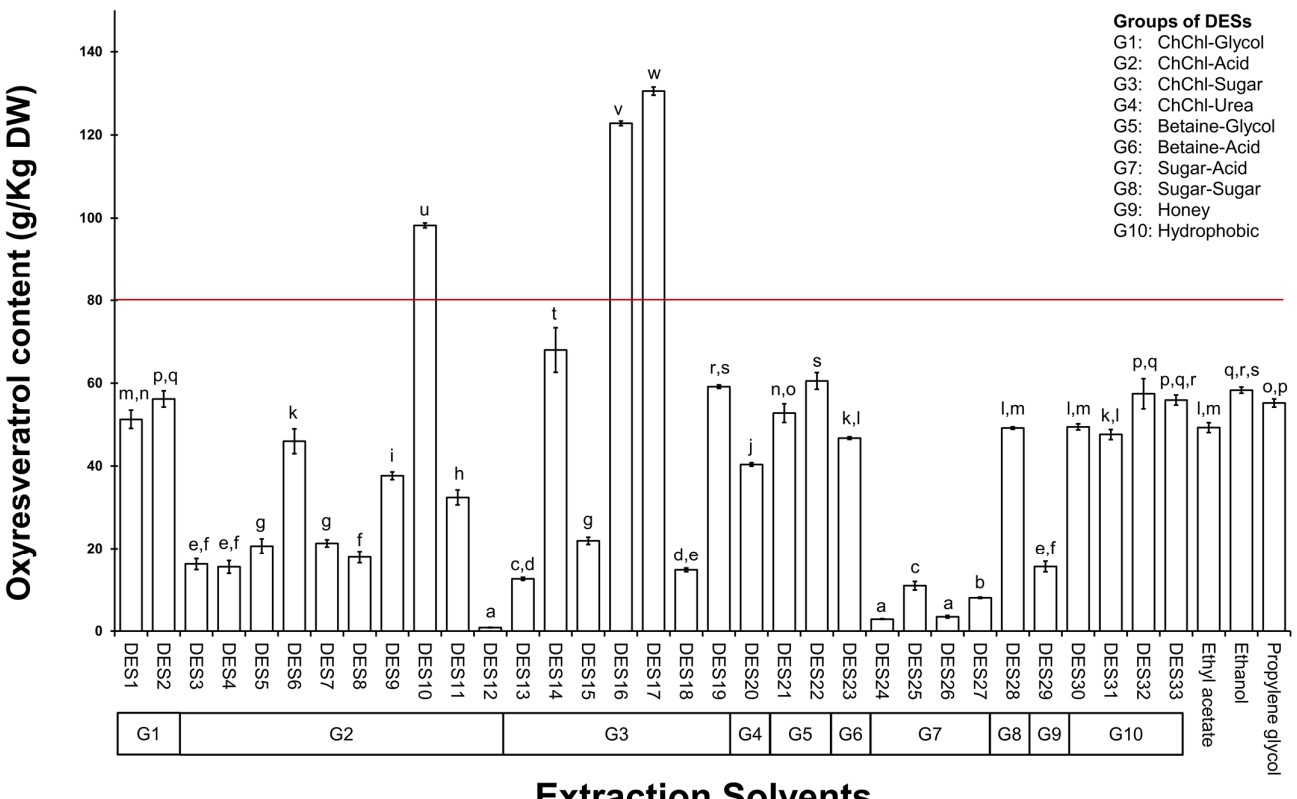

**Figure 4.** ORV content of AL extracts with different DESs and classical solvents, ethyl acetate, ethanol, and propylene glycol. The red line is the minimum criterion for oxyresveratrol extraction in this study. The different lowercase letters in the upper part of the bar chart (mean ± SD) represent statistically significant differences ($p < 0.05$) in ORV content with different extraction solvents (Tukey's post hoc test).

*3.4. Response Surface Optimization of Extraction*

3.4.1. Box–Behnken Design Fitting the Model

The BBD of RSM was used to define the optimum level of parameters that provided maximum ORV content and understand the relationship between extraction parameters and ORV yield. The ORV content of 15 runs of DES10 and DES17 is shown in Table 3, where X1, X2, and X3 are extraction parameters, time (min), temperature (°C), and water as a co-solvent (molar ratio), respectively. In this work, the results of 15 experiments were then theoretically optimized, and the optimal conditions were determined. The variance analysis of the quadratic models designed for ORV content (Y) of DES10 and DES17 for the selected quadratic prediction model suggested that the models were statistically significant ($p < 0.05$), as shown in Table 4, which also indicated the reliability of the model. ANOVA was employed to analyze a quadratic model and the fit of the response. The results are

shown in Table 4. The statistical significance of the model is usually determined by the $p$-value and the lack of fit (F-value), which is a diagnostic analysis for the adequacy of a model. The F-value results were 6.7 and 93.37 for Y (ORV) of DES10 and DES17, respectively, and a $p$-value of 0.0248 and <0.0001 for Y (ORV) of DES10 and DES17, respectively. The factors $X_2$, $X_1X_3$, $X_2X_3$, and $X_2^2$ for DES10 (Y) and $X_1$, $X_2$, $X_3$ for the model was $X_1^2$, $X_2^2$, $X_3^2$ and significantly affected the quadratic model, as shown in Table 4.

**Table 3.** BBD design and experimental results obtained for the measured responses with AL extract with DES10 and DES17.

| Exp. | Independent Variables | | | | Oxyresveratrol Contents (mg/kg DW) | |
| | $X_1$: Extraction Time (min) | $X_2$: Temperature (°C) | $X_3$: Molar Ratio of Water | | DES10 | DES17 |
| | | | DES10 | DES17 | | |
| 1 | 5.0 (−1) | 35.0 (−1) | 54.0 (0) | 32.5 (1) | 71.9246 | 50.4217 |
| 2 | 5.0 (−1) | 47.5 (0) | 3.0 (−1) | 0 (−1) | 80.1241 | 114.8703 |
| 3 | 5.0 (−1) | 47.5 (0) | 105.0 (1) | 65 (1) | 69.4065 | 77.4764 |
| 4 | 5.0 (−1) | 60.0 (1) | 54.0 (0) | 32.5 (1) | 61.4346 | 72.4134 |
| 5 | 62.5 (0) | 35.0 (−1) | 3.0 (−1) | 0 (−1) | 78.1610 | 106.1872 |
| 6 | 62.5 (0) | 35.0 (−1) | 105.0 (1) | 65 (1) | 71.5063 | 58.1077 |
| 7 | 62.5 (0) | 47.5 (0) | 54.0 (0) | 32.5 (1) | 79.4533 | 65.3372 |
| 8 | 62.5 (0) | 47.5 (0) | 54.0 (0) | 32.5 (1) | 71.2119 | 67.601 |
| 9 | 62.5 (0) | 47.5 (0) | 54.0 (0) | 32.5 (1) | 73.9160 | 69.8218 |
| 10 | 62.5 (0) | 60.0 (1) | 3.0 (−1) | 0 (−1) | 60.8940 | 109.6324 |
| 11 | 62.5 (0) | 60.0 (1) | 105.0 (1) | 65 (1) | 72.2889 | 68.1716 |
| 12 | 120.0 (1) | 35.0 (−1) | 54.0 (0) | 32.5 (1) | 70.0732 | 43.7554 |
| 13 | 120.0 (1) | 47.5 (0) | 3.0 (−1) | 0 (−1) | 69.3091 | 105.4751 |
| 14 | 120.0 (1) | 47.5 (0) | 105.0 (1) | 65 (1) | 77.1972 | 56.9957 |
| 15 | 120.0 (1) | 60.0 (1) | 54.0 (0) | 32.5 (1) | 59.0269 | 51.5038 |

**Table 4.** ANOVA results for quadratic model obtained from BBD.

| Term | Df | ORV Content | | | | | | | |
| | | DES10 | | | | DES17 | | | |
| | | Sum of Square | Mean Square | $F$-Value | $p$-Value | Sum of Square | Mean Square | $F$-Value | $p$-Value |
| Model | 9 | 553.05 | 61.45 | 6.7 | 0.0248 | 7617.88 | 846.43 | 93.37 | <0.0001 |
| $X_1$ | 1 | 6.63 | 6.63 | 0.7232 | 0.4339 | 346.69 | 346.69 | 38.24 | 0.0016 |
| $X_2$ | 1 | 180.7 | 180.7 | 19.71 | 0.0068 | 180.28 | 180.28 | 19.89 | 0.0066 |
| $X_3$ | 1 | 0.4563 | 0.4563 | 0.0498 | 0.8323 | 262.98 | 262.98 | 29.01 | 0.003 |
| $X_1X_2$ | 1 | 0.0774 | 0.0774 | 0.0084 | 0.9304 | 50.72 | 50.72 | 5.59 | 0.0643 |
| $X_1X_3$ | 1 | 86.54 | 86.54 | 9.44 | 0.0277 | 30.72 | 30.72 | 3.39 | 0.125 |
| $X_2X_3$ | 1 | 81.45 | 81.45 | 8.88 | 0.0308 | 10.95 | 10.95 | 1.21 | 0.3218 |
| $X_1^2$ | 1 | 32.67 | 32.67 | 3.56 | 0.1177 | 90.17 | 90.17 | 9.95 | 0.0253 |
| $X_2^2$ | 1 | 145.21 | 145.21 | 15.84 | 0.0105 | 243.53 | 243.53 | 26.86 | 0.0035 |
| $X_3^2$ | 1 | 16.65 | 16.65 | 1.82 | 0.2357 | 2507.42 | 2507.42 | 276.58 | <0.0001 |
| Residual | 5 | 45.84 | 9.17 | | | 45.33 | 9.07 | | |
| Lack of Fit | 3 | 10.54 | 3.51 | 0.1991 | 0.8897 | 35.27 | 11.76 | 2.34 | 0.3136 |
| Pure Error | 2 | 35.3 | 17.65 | | | 10.06 | 5.03 | | |
| Cor total | 14 | 598.9 | | | | 7663.21 | | | |
| $R^2$ | | 0.9235 | | | | 0.9941 | | | |
| Adj $R^2$ | | 0.7857 | | | | 0.9834 | | | |
| Pred $R^2$ | | 0.5857 | | | | 0.9234 | | | |
| Adequate precision | | 8.0597 | | | | 28.2882 | | | |

In the study, the linear regression with predicted $R^2$ was 0.9834 and 0.7857 for DES17 and DES10, respectively, and the adjusted $R^2$ was reasonably consistent with the adjusted $R^2$, with a deviation of less than 0.2, indicating that the models were reliable and accurate in predicting the response. In addition, the lack of fit value was 0.1991 ($p = 0.8897$) for DES10 and 2.34 ($p = 0.3136$) for DES17, indicating that the response fit the model.

### 3.4.2. Effect of the Extraction Variables on the Oxyresveratrol Content Using Box Behnken Design

The regression between the independent and dependent variables of ORV content, defined as Y, and the interaction factors as $X_1$, $X_2$, and $X_3$ for extraction time, temperature, and the molar ratio of water as a co-solvent, respectively, provides the code equations of the predicted model via the following Equations (4) and (5):

$$\text{DES10}: Y_{\text{Oxyresveratrol contents (g/kg DW)}} = 24.90272 + 0.02016X_1 + 3.0625X_2 - 0.518834X_3 - 0.000193X_1X_2 + 0.001586X_1X_3 + 0.007078X_2X_3 - 0.0009X_1{}^2 - 0.040135X_2{}^2 + 0.000816X_3{}^2 \tag{4}$$

$$\text{DES17}: Y_{\text{Oxyresveratrol contents (g/kg DW)}} = 374.340513 - 0.940644X_1 - 9.121504X_2 - 3.035763X_3 + 0.001513X_1X_2 + 0.008319X_1X_3 + 0.050797X_2X_3 + 0.005299X_1{}^2 + 0.071855X_2{}^2 - 0.004358X_3{}^2 \tag{5}$$

The effect of extraction time, extraction temperature, and ratio of water on ORV content using DES10 and DES17 as solvents was illustrated in 3D response surface graphs. The 3D response surface graphical representations are shown in Figure 5(a1–a3) for DES10 and Figure 5(b1–b3) for DES17. The 2D contour plots of DES10 and DES17 are presented in Figure S3(a1–a3) and Figure S3(b1–b3), respectively. The extraction efficiency peaks of DES10 (red range in 3D) suggest that lower water quantities, temperature, and time for extraction result in higher ORV content. The results showed that variations in temperature and extraction time had negligible effects on extraction efficiency, while a reduction in co-solvent (water) resulted in a higher ORV yield. This also supported the result of the DES10 model that a lower amount of water is required for a higher ORV content to avoid degradation during extraction. This intriguing result of the DES17 model suggests that the potential factors influencing ORV content are shorter extraction times and lower energy consumption at low temperatures.

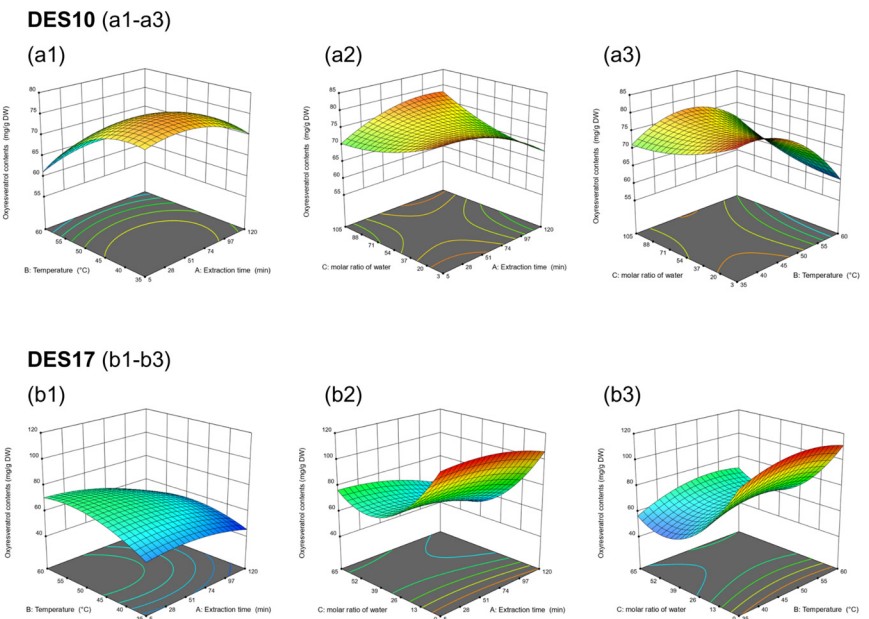

**Figure 5.** Three-dimensional response surface curve showing the influences of independent variables on DES10 (**a1–a3**) and DES17 (**b1–b3**). The colors red, orange, yellow, green, and blue represent the highest and lowest ORV content.

### 3.4.3. Verification of Predictive Model

Based on the interaction between the factors and the optimal condition for the ORV content of the quadratic model equations in predicting the optimal response values to maximize the ORV content, the best conditions for DES10 and DES17 were provided as follows:

DES10 had a temperature of 38 °C, an extraction time of 9.75 min, and a molar ratio of water in DES10 of 3, while DES17 had a temperature of 52 °C, an extraction time of 52 min, and a molar ratio of water in DES17 of 0. These conditions, which were identified as optimal via the RSM optimization approach, were further applied for experimental verification. The results are shown in Table 5. The ORV content of 82.672 and 115.714 mg/kg DW for DES10 and DES17 were expected from the models, with 95% prediction intervals of 73.99–91.34 and 109.07–122.35 mg/kg, respectively. The experimental yields of ORV were 81.225 ± 1.0167 and 110.485 ± 1.9072 mg/kg DW for DES10 and DES17, respectively. These results showed that the predicted accuracies of DES10 and DES17 were 98.78 and 95.65%, and their prediction errors were 1.75 and 4.52%, respectively. Thus, the model verification results confirmed the reliability of the model for the enrichment of ORV content from the AL, and the experiment was successfully optimized for the extraction condition with two solvents, DES10 and DES17.

**Table 5.** Comparative ORV content of AL extracts with DES10 and DES17.

| Responses | DES10 | DES17 |
|---|---|---|
| Mean of predicted value (ORV content; mg/kg DW) | 82.672 | 115.714 |
| Mean of experimental value (ORV content; mg/kg DW) | 81.225 ± 1.0167 | 110.485 ± 1.9072 |
| Error in relation to predicted value (%) | 1.75 | 4.52 |

### 3.5. Bioactivity Properties of Optimized Extracts

After optimization, an enhancement in bioactivity was expected. The ready-to-use extracts of DES10 and DES17 were prepared under optimal extraction conditions and evaluated for their biological activity in terms of antioxidant activity by DPPH assay and skin lightening by tyrosinase inhibition assay. The results are presented in Table 6 and reveal that the DES17 extract showed significantly stronger antioxidant activity compared to the DES10 extract and ascorbic acid, approximately 27-fold and 17-fold, respectively. As for antioxidation activity via the hydrogen transfer mechanism of DPPH radical, ORV is recognized for its antioxidation properties, and the result was consistent with a previous report [17]. Remarkably, DES17 extract showed higher inhibitory activity compared to DES10 extract and kojic acid, by approximately 3-fold and 252-fold, respectively. The biological test results of DES10 and DES17 corresponded to the ORV content in the extracts.

**Table 6.** DPPH radical scavenging activity and tyrosinase inhibition activity of optimized extracts ($n = 3$).

| Samples | IC$_{50}$ (µg/mL) | |
|---|---|---|
| | DPPH Radical Scavenging Activity | Tyrosinase Inhibition |
| DES10 | 8.72 ± 0.241 [a] | 0.41 ± 0.011 [b] |
| DES17 | 0.32 ± 0.004 [c] | 0.12 ± 0.003 [c] |
| Ascorbic acid | 5.46 ± 0.197 [b] | ND |
| Kojic acid | ND | 30.23 ± 0.122 [a] |

All analyses are means of triplicate measurements ± standard deviation. Means not sharing a common letter in columns were significantly different at $p < 0.05$. ND: not determined.

### 3.6. Scanning Electron Microscopy

The morphology of dried AL plant material was evaluated by using scanning electron microscopy (SEM) to validate the similarity and extraction efficiency of ready-to-use DES extracts, DES10 and DES17, compared to traditional organic solvents. Both the AL powder

before and after extraction with classical solvents (ethanol, ethyl acetate, and propylene glycol) were compared to those extracted with DES10 and DES17. The results revealed that the surface of dried AL powder displayed a rough texture before extraction (see Figure 6a,b), whereas smoother surfaces were observed after extraction with all tested solvents, including DES10 (see Figure 6c,d), DES17 (see Figure 6e,f), ethyl acetate (see Figure 6g,h), ethanol (see Figure 6i,j), and propylene glycol (see Figure 6k,l).

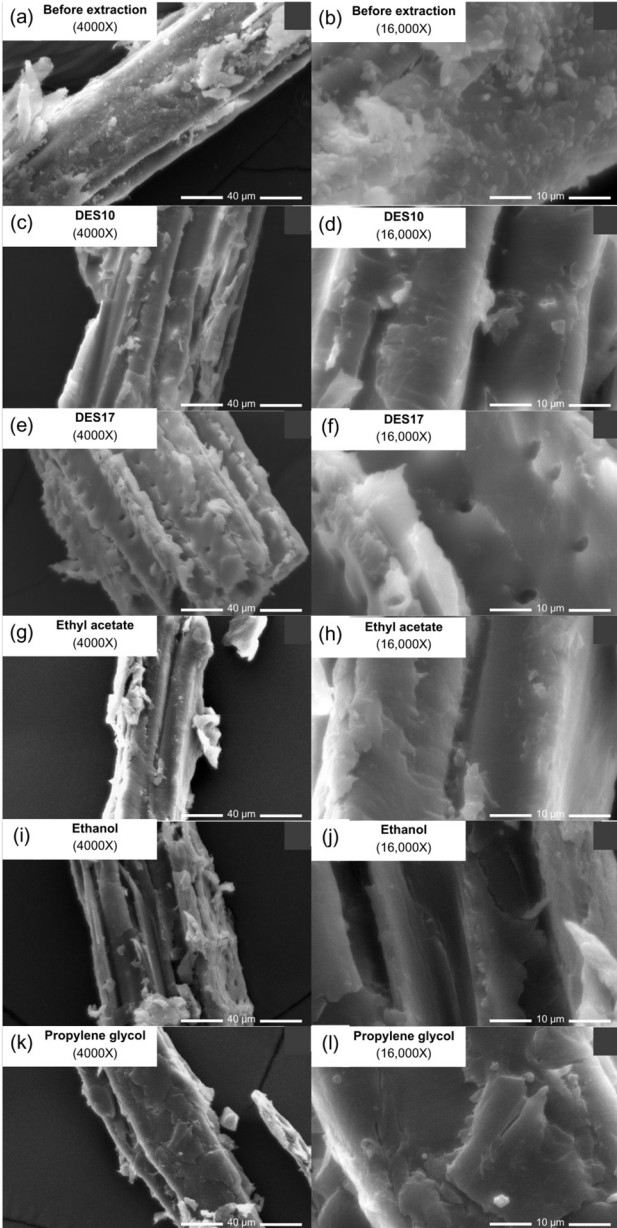

**Figure 6.** Morphology of AL before and after extraction with various solvents. SEM images of AL: before extraction, (**a**) 4000× and (**b**) 16,000×. After extraction with DES 10 (**c**) 4000× and (**d**) 16,000×; DES 17 (**e**) 4000× and (**f**) 16,000×; ethyl acetate (**g**) 4000× and (**h**) 16,000×; ethanol (**i**) 4000× and (**j**) 16,000×; and propylene glycol (**k**) 4000× and (**l**) 16,000×.

## 4. Discussion

The basic properties of DES, including water miscibility, pH, viscosity, and conductivity, play a crucial role in the selection of suitable solvents for the extraction of bioactive substances from plants. When selecting hydrophilic or hydrophobic DESs for plant extraction, hydrophilic DESs are suitable for polar to semi-polar bioactive compounds, while

hydrophobic DESs are effective for semi-polar to non-polar bioactive compounds in plants. The pH of DESs is a key factor influencing the stability and chemical interaction of bioactive compounds in plants [61,62] and the pH of DESs depends on its composition [63]. The viscosity of DESs was mostly low in the range of 20–100 cP, which was suitable for extraction as it diffused faster through the plant material, resulting in high extraction efficiency. The higher viscosity group (>100 Cp) was not suitable for industrial applicability [64]. The DES16 and DES17 had a high viscosity in 3576.00 ± 0.015 cP and 4505.00 ± 8.663 cP, respectively. However, to address this drawback of hydrophilic DESs, the addition of more water is necessary and applicable to industry [65], and in this study, the viscosity was optimized by adding water to high-viscosity DESs before extraction. The difference in viscosity could be due to the interaction of HBD and HBA with a widely branched network of hydrogen bonds, van der Waals forces, and electrostatic interactions between the different species [66]. The difference in electrical conductivity might be due to the ion size and viscosity of DESs [67]. The conductivity of DESs could be useful for the characterization of ionic interactions between HBD and HBA and for media polarity [68]. The HPLC assay was used for the analysis of ORV and validated prior to sample analysis. The robustness of the method was evaluated using intra-day and inter-day analyses, ensuring precision and accuracy of the assay was determined using the spiking technique. The validated HPLC method provided a short run time of only 7 min, and the results were in the acceptable criteria according to the ICH guidelines that the correlation coefficient (>0.9950), precision (%RSD < 5%), and recovery (80–120%) supported the suitability of the HPLC method for ORV analysis.

Commercially available AL extracts are predominantly sold in semi-solid form and are often produced using harsh and toxic solvents. The challenge in cosmetic formulation is the development of a less harmful extract and improved water solubility, a step that can be time-consuming. In response to this issue, "ready-to-use extracts" in liquid form have been developed to overcome the limitations of conventional extracts, which are typically diluted in solvents such as propylene glycol and ethanol, resulting in reduced bioactivity and efficacy. DESs have emerged as a promising solvent with an environmentally friendly, high extraction yield of bioactive compounds and the ability to produce liquid extracts [27,60,69]. In addition, DESs are also useful for industrial applications due to their simplicity in preparation, low volatility, biodegradability, low toxicity, and affordability [70,71]. This innovation not only streamlines the production of ready-to-use extracts but also enhances bioactivity. However, the high viscosity of DESs poses some limitations for the extraction process, as the diffusivity in the extraction of plant material is low, resulting in low extraction efficiency [72]. This obstacle can be overcome by applying external physical forces such as microwaves, high temperatures, or the addition of water or co-solvent [60,73]. In addition to the low volatility of DESs, which makes plant separation/isolation of bioactive compounds for further purification difficult, this suggests that DESs may be more suitable for extraction than plant isolation. Thirty-three DESs were screened for ORV extraction from AL. ChChl/citric acid/water (DES10), ChChl/sorbitol (DES16), and ChChl/xylose (DES17) showed high performance, yielding ORV content in the range of 100–130 mg ORV/kg DW, which were 2–3 times higher than those of classical solvents. These results were consistent with previous findings demonstrating the efficacy of ChChl/acid in flavonoid extraction [60]. However, there has been no report on ChChl/sugar for flavonoid extraction from plants. However, only two DESs that were representatives of G2 and G3 were selected for optimization of extraction conditions to maximize ORV.

The response surface methodology's BBD was employed to determine the optimal parameter levels for achieving maximum ORV content and study the relationship between extraction parameters and ORV yield. The optimal conditions and F-value outcomes for Y (ORV) in DES10 and DES17 were 6.7 and 93.37, respectively, with corresponding *p*-values of 0.0248 and <0.0001. The linear regression with predicted $R^2$ values of 0.9834 and 0.7857 for DES17 and DES10, respectively, indicated high reliability and accuracy in predicting the response. The adjusted $R^2$ values were reasonably consistent, deviating by less than

0.2. Furthermore, the lack of fit values for DES10 (0.1991, *p* = 0.8897) and DES17 (2.34, *p* = 0.3136) suggested that the response fit the model. The extraction model established in this study demonstrated that lower water quantities, temperature, and extraction time resulted in higher ORV content, resembling resveratrol. Previous research has highlighted the susceptibility of ORV to oxidation degradation when exposed to oxygen, light, and elevated temperatures [66]. It was the most stable at pH 3 to 8 [17]. Therefore, elevating the temperature and water levels in the extraction conditions may accelerate the degradation process of ORV during extraction, and prolonged exposure to these parameters tends to reduce ORV yield. It is worth noting that DES10 is characterized by an acidic environment with a pH of around 1.5, which could gradually impact ORV degradation when exposed for extended periods. This experiment has shown that a shorter time, a higher temperature, and a suitable pH value are required for the extraction of eutectic with a high ORV content. DES17, a member of the ChChl/sugar group, was applied for the first time to investigate the effects of the three factors of extraction temperature, extraction time, and amount of water on the ORV content of AL. In addition, the ORV content in DES17 appeared to be more stable and yielding than in DES10, probably due to its pH of about 3.8. This study points out the advantages of RSM in improving ORV content using DESs, reducing the cost of the optimization process, and providing the optimal conditions for extraction.

The optimal conditions for maximum ORV extraction of DES10 and DES17 were provided. The predicted accuracies of DES10 and DES17 were 98.78 and 95.65%, and their prediction errors were only 1.75 and 4.52%, respectively. Therefore, the model verification results affirmed the reliability of the model in enriching ORV content from the AL, and the extraction conditions for two solvents, DES10 and DES17, were successfully optimized.

Remarkably, DES17 extract showed higher tyrosinase inhibitory and antioxidant activity than DES10 extract and kojic acid, by approximately 3-fold and 252-fold, and 17-fold and 27-fold for ascorbic acid and DES10. These were consistent with the ORV content in the DES extracts. This study pointed out that the oxyresveratrol-enriched extract of DES17 has outstanding biological activity, which is stronger than that of the positive controls for antioxidant and tyrosinase inhibition, and that the ready-to-use liquid extract simultaneously provides for more convenient use in cosmetic formulations. Interestingly, the dried plant material extracted with ethyl acetate and ethanol exhibited some porousness on the surface, possibly due to the corrosive effects of these harsh solvents, which were not observed with DESs. This observation confirms the comparable effectiveness of DES for extraction to that of conventional solvents.

## 5. Conclusions

In this research, we successfully developed a ready-to-use DES system that shows the potential to extract high concentrations of ORV via an environmentally friendly extraction process. We thoroughly evaluated 33 DESs to determine the optimal conditions for maximizing ORV content under specific extraction parameters. Three DESs from this screening: ChChl/citric acid/water (DES10) in the ChChl/acid group (G2), ChChl/sorbitol (DES16), and ChChl/xylose (DES17) in the ChChl/sugar group (G3), exhibited outstanding ORV yield in the range of 100–130 mg/kg DW. Two promising DESs, DES10 and DES17, were chosen as representatives of ChChl/acid (G2) and ChChl/sugar (G3) members, respectively, for optimization of extraction conditions to maximize ORV content using the BBD of the RSM. The predicted optimal conditions that gave the highest yield were experimentally validated as follows: for DES10, extraction at 38 °C for 9.75 min, the molar ratio of water in DES10 was 3, and for DES17, extraction at 52 °C for 52 min without the addition of water. The experimental results were largely consistent with the predicted results in terms of ORV content. Our ready-to-use DES extracts of DES10 and DES17 showed robust biological activities, especially in terms of antioxidant and anti-tyrosinase effects. Notably, the DES17 extract showed 17-fold and 252-fold higher antioxidant and anti-tyrosinase activities than standard ascorbic acid and kojic acid, respectively. This research represents a valuable

contribution to the cosmetic and pharmaceutical industries by offering a new generation of environmentally friendly extraction methods.

**Supplementary Materials:** The following supporting information can be downloaded at: https://www.mdpi.com/article/10.3390/cosmetics11020058/s1. Figure S1. Physicochemical properties of DESs were shown as the pH value (S1a), viscosity (S1b), and conductivity (S1c) of DESs; Figure S2. Calibration curves of ORV in the concentration range of 1–20 μg/mL; Figure S3. Contour plots showing the influences of independent variables on DES10 (S3a1–a3) and DES17 (S3b1–b3); Oxyresveratrol characterization session including Figures S4–S8; Figure S4 showed the chemical structure of oxyresveratrol, the Figure S5 exhibited the HPLC chromatogram of isolated oxyresveratrol with purity analysis. The $^1$H-NMR spectrum of oxyresveratrol (Acetone-d6, 400 MHz) is presented in Figure S6a,b, while the IR spectrum is shown in Figure S7. The mass spectrum of oxyresveratrol (ESI-MS with negative mode) is exhibited in Figure S8. The characterization information of the isolated compound was confirmed by all spectra as oxyresveratrol, which was consistent with previous report by Arriffin, N. et al. [74].

**Author Contributions:** Conceptualization, J.S., K.S. and N.K.; methodology, J.S., K.S., P.T., W.P. (Wipawan Prompan), W.P. (Warakhim Punan) and T.R.; writing original draft preparation, J.S. and K.S.; review and editing, J.S., K.S., N.K., P.T., W.P. (Wipawan Prompan), W.P. (Warakhim Punan), T.R., W.J. and P.T.-o.; resource, W.J.; funding acquisition, J.S. All authors have read and agreed to the published version of the manuscript.

**Funding:** This research was funded by (1) the Office of the Permanent Secretary, Ministry of Higher Education, Science, Research, and Innovation Fund (Grant No. RGNS 63-146); (2) the University of Phayao (UP), FF66 (FF66-RIM040), and FF65 (Grant No. 2377120 FF65-RIM098); and (3) the Department of Thai traditional and alternative medicine, Thai traditional medicine knowledge (Grant No. 3).

**Institutional Review Board Statement:** Not applicable.

**Informed Consent Statement:** Not applicable.

**Data Availability Statement:** Data are contained within the article and Supplementary Materials.

**Acknowledgments:** The authors thank the Research and Innovation Center in Cosmetic Science and Natural Products (RCCN) and the School of Pharmaceutical Sciences, University of Phayao, for their technical support. The authors would like to thank Nitra Nuengchamnong, Nungruthai Suphrom, and Ruttanaporn Chantakul, who supported MS and NMR analysis.

**Conflicts of Interest:** Author Wasinee Juprasert is an employee of Phitsanuchemical Co., Ltd. The authors declare that the research was conducted in the absence of any commercial or financial relationships that could be construed as a potential conflict of interest. The company had no role in the design of the study; in the collection, analyses, or interpretation of data; in the writing of the manuscript; or in the decision to publish the results.

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
