# Peer review of "Development of a Ready-to-Use Oxyresveratrol-Enriched Extract from Artocarpus lakoocha Roxb. Using Greener Solvents and Deep Eutectic Solvents for a Whitening Agent"

_cosmetics, doi:10.3390/cosmetics11020058_

Round 1
Reviewer 1 Report
Comments and Suggestions for Authors
The manuscript is interesting. However, I have some comments and suggestions for the authors.
1. Abstract: Is the value of 27-fold for DES17 activity for ascorbic acid correct? Should it not be 17-fold?
2. Line 11: It was stirred until a clear liquid was obtained. What was the average mixing time? Were they minutes or hours?
3. It was mentioned that the eutectic solvent consists of a hydrogen bond acceptor, such as choline chloride and betaine, and a hydrogen bond donor, such as citric acid, malic acid and xylitol. What about others that are listed in the table, such as, for example, fructose, tartaric acid, etc., and are not listed in lines 97 and 98?
4. I think that Table 1 should be in section 2. Materials and Methods.
5. Line 107: Please explain the abbreviation “DI”.
6. What was the column particle size?
7. Line 145:Please add reference number to “ICH guidelines (2022)”.
8. Why was the analytical standard for oxyresveratrol, which is commercially available with a certificate, not used for validation, but synthesised when the yield of oxyresveratrol synthesis was only 10%.
9. NMR, MS spectra data for isolated oxyresveratrol should be in the Supplementary Material.
10. Line 225: It should be “per kilogram”.
11. Line 240-241:”Choline chloride” instead of “chlorine”.
12. Table 3: “G3;ChChl-sugar” instead of “G3;ChCl-sugar”.
13. Line 271 and 272: Use the abbreviation or the full name. The abbreviation LOD and LOQ were explain in subtitle 2.4.
14. Line 436, 455: Abbreviation DES, ORV were explain in the Introduction.
15. Line 479: Use the abbreviation or the full name. The abbreviation BBD was explain in subtitle 2.6.
16. Line 480: Duplicate “The Response Surface Methodology's Box-Behnken Design (BBD) was employed to“.
17. Line 539: It should be:17-fold and 252-fold.
Author Response
Thank you for assessing our manuscript: We now act on or respond to your comments in RED:
REVIEWER 1
- Abstract: Is the value of 27-fold for DES17 activity for ascorbic acid, correct? Should it not be 17-fold?
We have corrected according to your advice. (Page 1, Line 29)
- Line 11: It was stirred until a clear liquid was obtained.What was the average mixing time? Were they minutes or hours?
We have added more details “The mixture was heated at ~80°C with constant stirring using a magnetic stirrer, until it became a clear liquid (approximately 1-2 hours).” (Page3, Line 112-114)
- It was mentioned that the eutectic solvent consists of a hydrogen bond acceptor, such as choline chloride and betaine, and a hydrogen bond donor, such as citric acid, malic acid, and xylitol. What about others that are listed in the table, such as, for example, fructose, tartaric acid, etc., and are not listed in lines 97 and 98?
We have added more details in Page3, Line 106-111 “The eutectic solvent consists of a hydrogen bond acceptor including choline chloride, betaine, citric acid, tartaric acid, malic acid, fructose, camphor, menthol, lauric acid and lactic acid and a hydrogen bond donor including ethylene glycol, 1,3-propanediol, lactic acid, malic acid, citric acid, glycolic acid, oxalic acid, p-toluene sulfonic acid, glucose, maltose, fructose, sorbitol, xylose, xylitol, urea, 1,3 propanediol, 1,2-butanediol, tartaric acid, glucose, thymol and menthol.”
- I think that Table 1 should be in section 2. Materials and Methods.
We have changed according to your suggestion. (Page3, Line 127)
- Line 107: Please explain the abbreviation “DI”.
We have added more detail as “Deionized water” in Page 3, Line 120.
- What was the column particle size?
We have added more details as “a 5 µm C-18(2) column (Phenomenex) with a 250 mm x 4.6 mm diameter” in Page4, Line 148.
- Line 145: Please add reference number to “ICH guidelines (2022)”.
We have added reference in Page 5, Line 160.
- Why was the analytical standard for oxyresveratrol, which is commercially available with a certificate, not used for validation, but synthesised when the yield of oxyresveratrol synthesis was only 10%.
This experiment proved that 10% of the oxyresveratrol actually comes from this plant. We used the isolated oxyresveratrol from the experiment as a reference standard, as it had a higher purity of 99% (HPLC analysis) than commercial oxyresveratrol, which was only 97% (HPLC analysis). We have also added in supplementary data on the purity of oxyresveratrol from the isolation experiment.
- NMR, MS spectra data for isolated oxyresveratrol should be in the Supplementary Material.
We have added NMR, IR, MS spectra data in the Supplementary data.
- Line 225: It should be “per kilogram”.
We have edited. (Page 6, Line 239)
- Line 240-241:”Choline chloride” instead of “chlorine”.
We have edited according to your suggestion. (Page 7, Line 254)
- Table 3: “G3;ChChl-sugar” instead of “G3;ChCl-sugar”.
We have edited according to your suggestion. (Table 3)
- Line 271 and 272: Use the abbreviation or the full name. The abbreviation LOD and LOQ were explain in subtitle 2.4.
We have edited according to your suggestion in Page 7, Line 285.
- Line 436, 455: Abbreviation DES, ORV were explained in the Introduction.
We have edited in Page 15, Line 445, 464.
- Line 479: Use the abbreviation or the full name. The abbreviation BBD was explained in subtitle 2.6.
We have edited in Page 15, Line 495.
- Line 480: Duplicate “The Response Surface Methodology's Box-Behnken Design (BBD) was employed to“.
We have edited in Page 15, Line 495.
- Line 539: It should be:17-fold and 252-fold.
We have corrected in Page 17, Line 554.

Reviewer 2 Report
Comments and Suggestions for Authors
General assessment: This innovative study introduces green technology using deep eutectic solvents (DESs) for Oxyresveratrol (ORV) extraction. It is known that there’s a high demand for ORV in the cosmetic and pharmaceutical industries. The Artocapus lakoocha Roxb. (AL) extract is important in dietary supplements and cosmetics due to its anti-tyrosinase and antioxidant properties. Two obtained DESs (DES10 and DES17) showed higher ORV content than conventional solvents. The main advantage is that the ORV-enriched extracts can be directly incorporated into cosmetic formulations without needing stock solutions, making them ready to use.
1.The title should be revised because it does not convey the essence of the study: the use of selected DESs for the efficient extraction of ORV. The words „as whitening agent” are not necessary in the title.
2.In the Introduction section, there is no reference to resveratrol and the known differences between it and oxyresveratrol (is it more advantageous?).
3.A table with examples of DESs used to date and active principles would be helpful in the Introduction section.
4.The term synthesis is less appropriate for obtaining DESs (lack of chemical reactions). I suggest replacing it with obtaining or preparing.
5.The advantages of using DESs can be better highlighted and structured during discussions.
6.In addition to the advantages, for a fair balance, it is necessary to emphasize the disadvantages of using DES, if they exist (costs, equipment, qualified personnel?).
7.The legend of the Table 1 is too brief.
8.In Figure 4, only the distinctive number of the DES is sufficient because it also includes a legend of G1-G10.
9.Abbreviations in headlines should be avoided (e.g. 3.4.1.)
10.Table 5 has no units of measure.
11.Line 49-50: Please, correct the word „Oxyreveratrol.”
Author Response
Thank you for assessing our manuscript: We now act on or respond to your comments in RED:
REVIEWER 2
(DESs) for Oxyresveratrol (ORV) extraction. It is known that there’s a high demand for ORV in the cosmetic and pharmaceutical industries. The Artocapus lakoocha Roxb. (AL) extract is important in dietary supplements and cosmetics due to its anti-tyrosinase and antioxidant properties. Two obtained DESs (DES10 and DES17) showed higher ORV content than conventional solvents. The main advantage is that the ORV-enriched extracts can be directly incorporated into cosmetic formulations without needing stock solutions, making them ready to use.
1.The title should be revised because it does not convey the essence of the study: the use of selected DESs for the efficient extraction of ORV. The words, as whitening agent” are not necessary in the title.
In this project, the oxyresveratrol extraction efficiency of 33 DESs, that categorized into 10 classes including G1: ChChl-glycol, G2: ChChl-acid, G3: ChChl-sugar, G4: ChChl-urea, G5: betaine-glycol, G6: betaine-acid, G7: sugar-acid, G8: sugar-sugar, G9: natural DES, and G10: hydrophobic DESs were comprehensively investigated in the section 3.3, and the whitening properties were studied in the experiment with the tyrosinase inhibition assay in the Section 2.8. For this reason, we have changed the title to “Development of ready-to-use oxyresveratrol-enriched extract from Artocarpus lakoocha Roxb. using greener solvents, deep eutectic solvents for whitening agent”
2.In the Introduction section, there is no reference to resveratrol and the known differences between it and oxyresveratrol (is it more advantageous?).
We have edited and added more detail according to reviewer’s suggestion as “In addition, bioactive compounds that have skin-whitening properties in AL are mainly found as oxyresveratrol (ORV) and resveratrol, according to recent research reports [16]. When comparing the skin whitening properties of ORV, it was found that the inhibition of tyrosinase or melanogensis of oxyresveratrol has better efficiency compared to resveratrol” (Page1-2, Line 49-53).
3.A table with examples of DESs used to date and active principles would be helpful in the Introduction section.
We have added more detail as “A mixture of two or more pure chemicals that, when mixed in the appropriate ratio, results in a eutectic mixture that deviates from ideal thermodynamic behavior is known as a DESs. Strong interactions between the initial components that function as hydrogen bond donors (HBDs) such as citric acid, malic acid and hydrogen bond acceptors (HBAs) such as choline chloride and betaine. DESs have been proven to have a wide range of applications, including synthesis, extraction, biocatalysis, nanomaterials, biotechnology, electrochemistry, food, cosmetics, drugs or biofuel.” (Page 2, Line 64-70)
4.The term synthesis is less appropriate for obtaining DESs (lack of chemical reactions). I suggest replacing it with obtaining or preparing.
We have changed to be “preparation”.
5.The advantages of using DESs can be better highlighted and structured during discussions.
The advantages of DES have added in discussion part in Page 15, Line 478-480.
6.In addition to the advantages, for a fair balance, it is necessary to emphasize the disadvantages of using DES, if they exist (costs, equipment, qualified personnel?).
We have added more detail in Page 15, Line 481-487.
7.The legend of the Table 1 is too brief.
We have edited according to the suggestion as “Groups, components, and solubility property of DESs in this study”. (Table1)
8.In Figure 4, only the distinctive number of the DES is sufficient because it also includes a legend of G1-G10.
We have edited according to your suggestions.
- Abbreviations in headlines should be avoided (e.g. 3.4.1.)
We have edited and revised through manuscript.
10.Table 5 has no units of measure.
We have edited.
11.Line 49-50: Please, correct the word „Oxyreveratrol.”
We have edited. (Page 2, Line 54)
